# Survival of glioblastoma treated with a moderately escalated radiation dose—Results of a retrospective analysis

**Li-Tsun Shieh[1], How-Ran Guo[2,3], Chung-Han Ho [4,5], Li-Ching Lin[6], Chin-Hong Chang[7], Sheng-Yow Ho [1,6,8]\***

1 Department of Radiation Oncology, Chi Mei Medical Center, Liouying, Tainan, Taiwan, Republic of china, 2 Department of Environmental and Occupational Health, College of Medicine, National Cheng Kung University, Tainan, Taiwan, Republic of china, 3 Department of Occupational and Environmental Medicine, National Cheng Kung University Hospital, Tainan, Taiwan, Republic of china, 4 Department of Medical Research, Chi Mei Medical Center, Tainan, Taiwan, Republic of china, 5 Department of Hospital and Health Care Administration, Chia Nan University of Pharmacy and Science, Tainan, Taiwan, Republic of china, 6 Department of Radiation Oncology, Chi Mei Medical Center, Tainan, Taiwan, Republic of china, 7 Department of Neurosurgery, Chi Mei Medical Center, Tainan, Taiwan, Republic of china, 8 Graduate Institute of Medical Science, Chang Jung Christian University, Tainan, Taiwan, Republic of china

\* shengho@seed.net.tw

**Data Availability Statement:** All relevant data are within the paper and its Supporting Information files.

## Abstract

Glioblastoma (GBM) has the highest fatality rate among primary malignant brain tumors and typically tends to recur locally just adjacent to the original tumor site following surgical resection and adjuvant radiotherapy. We conducted a study to evaluate the survival outcomes between a standard dose ($\leq$ 60 Gy) and moderate radiation dose escalation (>60 Gy), and to identify prognostic factors for GBM. We retrospectively reviewed the medical records of primary GBM patients diagnosed between 2005 and 2016 in two referral hospitals in Taiwan. They were identified from the cancer registry database and followed up from the date of diagnosis to October 2018. The progression-free survival (PFS) and overall survival (OS) were compared between the two dose groups, and independent factors for survival were analyzed through Cox proportional hazard model. We also affirmed the results using Cox regression with least absolute shrinkage and selection operator (LASSO) approach. From our cancer registry database, 142 GBM patients were identified, and 84 of them fit the inclusion criteria. Of the 84 patients, 52 (62%) were males. The radiation dose ranged from 50.0 Gy to 66.6 Gy, but their treatment volumes were similar to the others. Fifteen (18%) patients received an escalated dose boost >60.0 Gy. The escalated group had a longer median PFS (15.4 *vs.* 7.9 months, *p* = 0.01 for log-rank test), and a longer median OS was also longer in the escalation group (33.8 *vs.* 12.5 months, *p* <0.001) than the reference group. Following a multivariate analysis, the escalated dose was identified as a significant predictor for good prognosis (PFS: hazard ratio [HR] = 0.48, 95% confidence interval [95%CI]: 0.23–0.98; OS: HR = 0.40, 95%CI: 0.21–0.78). Using the LASSO approach, we found age > 70 (HR = 1.55), diagnosis after 2010 (HR = 1.42), and a larger radiation volume ($\geq$ 250ml; HR = 0.81) were predictors of PFS. The escalated dose (HR = 0.47) and a larger radiation volume (HR = 0.76) were identified as predictors for better OS. Following detailed statistical analysis, a moderate radiation dose escalation (> 60 Gy) was found as an independent factor affecting

**Funding:** SYH. This study was supported by the grant of Chi Mei Medical Center, Liouying (No. CLFH10840). The funders had no role in study design, data collection and analysis, decision to publish, or preparation of the manuscript.

**Competing interests:** The authors have declared that no competing interests exist.

**Abbreviations:** 3D CRT, 3D conformal radiotherapy; AICC, Akaike information criterion with a correction; CI, confidence interval; CTV, clinical target volume; GBM, Glioblastoma; GTR, gross tumor resection; GTV, gross tumor volume; HR, hazard ratio; LASSO, least absolute shrinkage and selection operator; OS, overall survival; PFS, progression-free survival; RTOG, radiation therapy and oncology group; SIB, simultaneous integrated boost; TMZ, temozolomide.

OS in GBM patients. In conclusion, a moderate radiation dose escalation (> 60 Gy) was an independent predictor for longer OS in GBM patients. However, prospective studies including more patients with more information, such as molecular markers and completeness of resection, are needed to confirm our findings.

## Introduction

Glioblastoma (GBM) is the most common central nervous system tumor in adults and the deadliest primary malignant brain tumor. Patients succumb to death shortly after diagnosis, despite multi-modal therapy. Surgical resection of the tumor followed by concurrent chemo-radiotherapy is the standard treatment nowadays, and management of GBM has advanced little in the last two decades. The overall 2-year survival of GBM following optimal treatment is only 27% [1, 2]. GBM typically occurs in a single brain lesion and tends to recur locally just adjacent to the original tumor site following resection and adjuvant radiotherapy. According to previous studies on the pattern of failure for GBM patients, more than 80% of tumor relapse occurred within the initial tumor margin [3–5]. Given the importance of local control, it is advised to achieve gross tumor resection (GTR) and administer an escalated radiation dose in order to avoid local tumor recurrence. An analysis of the National Cancer Database revealed that patients who received GTR did have an increased survival rate over patients who underwent subtotal resection or biopsy only [6]. However, completion of GTR would increase the risk of neurologic deficits.

In the era of conformal and precise radiation, several studies tried to administer an escalated radiation dose to the GBM tumor surgical bed or residuum to improve local control [5, 7–12]. Milano et al. investigated the pattern and timing of GBM treated by biopsy or resection followed by standard 60-Gy radiation and temozolomide (TMZ) therapy and found that 80% of recurrence still occurred in the primary tumor site [7]. Several previous studies administered an escalated radiation dose (> 60 Gy), even up to 90 Gy) with the aim to achieve better local control [5, 8–10]. Most of them failed to prolong survival through an escalated dose. However, two recent reports showed a positive prognostic effect of a moderately integrated radiation boost dose (> 60 Gy) with a superior local control, and this also translated to a better overall survival (OS) [11, 12].

Given the inconsistent clinical results, this study aimed to compare survival outcomes between the escalated radiation dose (> 60 Gy) and the standard radiation dose (≤ 60 Gy) in GBM patients undergoing post-operative adjuvant therapy. Prognostic factors associated with the survival of GBM were also evaluated.

## Materials and methods

### Study population

We retrospectively reviewed the medical records of primary brain GBM patients in a medical center and its affiliated branch, both in Tainan, Taiwan. Patients with histology codes ICD-O-3 9440/3 in the cancer registry database who were diagnosed between 2005 and 2016 were identified. GBM variants of gliosarcoma, giant-cell, or epithelioid histologic differentiation were excluded. The inclusion criteria were older than 20 years of age, good performance status (ECOG less than 2), and undergoing curative brain irradiation. We excluded patients who had no pathologic confirmation or had received brain irradiation before the diagnosis of GBM.

We extracted the following information from the database: age, sex, histology, extents of surgery, radiation dose, prescription of chemotherapy, and the date of last follow-up. The follow-up time was from the date of GBM diagnosis to October 2018.

The extent of surgery was assessed by both post-operative imaging and operation records. All the brain images were reviewed by an oncologist and a radiologist to confirm the size and location of the brain lesion. All patients received radiation using the technique of intensity-modulated radiation therapy (IMRT). The radiation dose 45–46 Gy was prescribed in 1.8–2.0 Gy daily fractions to the first planning target volume (PTV), which was defined as the gross tumor volume (GTV) plus perifocal edema for possible microscopic extension. The clinical target volume (CTV) was defined as GTV plus a 1.5–2.0 cm margin for microscopic extension, and the anatomical structure was considered. Then a subsequent boost of 15–20 Gy was given to GTV plus a 0.5–1.0 cm margin. The dose of the boost was determined on the basis of the physician's consideration. All patients were followed up with brain images and clinical visits regularly after therapy.

## Literature search strategy

Literature search was conducted in December, 2019. We searched PubMed (National Library of Medicine) between January 1995 and December 2019 using "glioblastoma" or "brain tumor," "dose escalation" or "higher dose," and "radiation" or "radiotherapy" as key words. We included human studies published in full-text English. Because we were interested in understanding the clinical studies of escalated radiation, articles were not reviewed if they were case reports, animal studies, reports of radiation secondary to other conditions, or reports of surgical or radiological management of GBMs.

## Statistical analysis

OS and progression-free survival (PFS) were assessed in both groups. OS censored patients at death or the date of last follow-up, and PFS ended at locoregional recurrence or death from any cause. We applied Kaplan-Meier analyses for comparisons of survival and evaluated differences between the survival curves by the log-rank test. Differences in continuous variables were evaluated by the student $t$ test or Wilcoxon rank sum test, and those in categorical variables were evaluated by the Fisher's exact test. To identify independent factors for survival, Cox proportional hazards model that included factors with $p$ values smaller than 0.10 in the univariate analyses were constructed. In order to have better identification of survival factors, we further adopt Cox regression models with least absolute shrinkage and selection operator (LASSO) approach to establish the final model on the basis of Akaike information criterion with a correction (AICC) for small sample size [13–14]. Hazard ratios (HRs) and associated 95% confidence intervals (CIs) were calculated. A two-tailed $p$ value $< 0.05$ was considered statistically significant. All analyses were conducted by SPSS 24 for Windows (IBM, Armonk, NY, USA). SAS 9.4 for Windows software (SAS Institute, Cary, NC, USA) was used to apply the LASSO approach, with the SAS syntax PROC HPGENSELECT.

## Ethics statement

The retrospective analysis was approved by the Chi Mei Medical Center institutional review board (IRB) and ethics committee (IRB number: 10710-L05 and 10810-L05). All the methods were carried out in accordance with the approved guideline, and all data were fully anonymized before accessed. Written informed consent of the patients or their families was not judged necessary for this kind of retrospective study by the Chi Mei Medical Center IRB.

## Results

From 2005 through 2016, 142 GBM newly diagnosed patients were identified in our database of cancer registry, and 84 of them fit in inclusion criteria. Of the 58 ineligible GBM patients, 52 did not undergo radiotherapy, 4 received palliative brain irradiation (< 50 Gy), and 2 were younger than 20 years old. The patient's age at diagnosis ranged from 21 to 84 years old (median: 61 years) (Table 1). Of the 84 patients, 52 (62%) were males, and 27 (32%) were diagnosed before 2010. Most (63, 75%) of the patients underwent subtotal or GTR, and 69 (82%) received chemotherapy. Among the patients who received chemotherapy, 46 (67% of the total) were prescribed with a TMZ-based regimen. The radiation dose ranged from 50.0 Gy to 66.6 Gy with a median of 60.0 Gy. While 15 (18%) patients received dose escalation boost > 60.0 Gy with a median of 66 Gy (range: 61.2 to 66.6 Gy), only 4 patients did not reach the full 66-Gy boost dose, but no explicit medical records could be obtained. The irradiation volume ranged from 56 to 817 mL (median: 247 mL). The median OS of all enrolled patients was 14.0 months, while the median PFS was only 9.0 months.

The 15 patients with dose-escalation (> 60 Gy) (escalated group) were younger than the 69 patients who received the standard 60-Gy dose (reference group) (median, 52 *vs.* 62 years old, $p = 0.05$), but the differences in age, sex, surgery, and chemotherapy did not reach statistical significance (Table 2). The median radiation treatment volume was similar in both groups (249 ml in dose escalation *vs.* 247 ml in standard dose). Following age stratification at 70 years, we found patients older than 70 years seemed to be more likely to receive the standard dose,

**Table 1. Patient characteristics of glioblastoma.**

| Characteristics | Patients (N = 84) (n, %) |
|---|---|
| Age (year) (median, range) | 61 (2–84) |
| Overall survival (month) (median, range) | 14.0 (2.5–109.8) |
| Progression-free survival (month) (median, range) | 9.0 (1.4–61.4) |
| Sex | |
| Male | 52 (62) |
| Female | 32 (38) |
| Operation[a] | |
| Yes | 63 (75) |
| Biopsy | 21 (25) |
| Radiation dose (Gy) (mean, range) | 60.0 (50.0–66.6) |
| > 60 | 15 (18) |
| ≤ 60 | 69 (82) |
| Radiation volume (mL) (median, range)[b] | 247 (56–817) |
| Chemotherapy | |
| Yes | 69 (82) |
| No | 15 (18) |
| Temozolomide-based regimen | |
| Yes | 46 (67) |
| Non-temozolomide regimen | 23 (33) |
| Year of diagnosis | |
| 2005–2010 | 27 (32) |
| 2010–2016 | 57 (68) |

[a]Subtotal or gross-total resection, other than biopsy only.
[b]27 patients did not have detailed radiation volume.

**Table 2. Comparison of glioblastoma patients treated with dose escalation (> 60 Gy) and standard dose (≤ 60 Gy).**

| Characteristics | Escalation (N, %) | Standard (N, %) | p value[a] |
|---|---|---|---|
| Patients number | 15 | 69 | |
| Overall survival (month) (median, range) | 33.8 (6.2–109.8) | 12.5 (2.5–77.2) | < 0.001 |
| Progression-free survival (month) (median, range) | 15.4 (13.1–61.4) | 7.7 (1.4–55.0) | 0.01 |
| Age (years) (median, range) | 52 (29–77) | 62 (21–84) | 0.05 |
| Age group (year) | | | 0.17 |
| > 70 | 1 (7) | 18 (26) | |
| ≤ 70 | 14 (93) | 51 (74) | |
| Sex | | | 0.24 |
| Male | 7 (47) | 45 (65) | |
| Female | 8 (53) | 24 (35) | |
| Surgery[b] | | | 0.51 |
| Yes | 10 (67) | 53 (77) | |
| Biopsy | 5 (33) | 16 (23) | |
| Chemotherapy | | | 0.73 |
| Yes | 12 (80) | 57 (83) | |
| No | 3 (20) | 12 (17) | |
| Temozolomide-based regimen[c] | | | 0.20 |
| Yes | 5 (56) | 41 (69) | |
| Non-temozolomide regimen | 4 (44) | 13 (31) | |
| Year of diagnosis | | | <0.01 |
| 2005–2010 | 11 (73) | 16 (23) | |
| 2010–2016 | 4 (27) | 53 (78) | |
| Radiation volume (mL) (median, range)[d] | 247 (94–817) | 249 (56–731) | 0.66 |

[a]p value for student t test, Wilcoxon rank sum test or Fisher's exact test.

[b]Subtotal or gross-total resection, other than biopsy only.

[c]6 patients did not have detailed records of chemotherapy regimen.

[d]27 patients did not have detailed records of radiation volume.

but the difference did not reach statistical significance (26% *vs.* 7%, $p = 0.17$). Following stratification by the year of diagnosis, we found a higher percentage of patients were assigned to receive the escalated dose before 2010 (73% vs. 27%, $p <0.01$).

The escalated group had a better PFS than the reference group, with a median of 15.4 (range: 13.1–61.4) *vs.* 7.9 (range: 1.4–55) months ($p = 0.01$ for log-rank test) (Fig 1). In addition, the escalated group survived longer, with a median OS of 33.8 months (range: 6.2–109.8) *vs.* 12.5 (range: 2.5–77.2) months ($p < 0.01$ for log-rank test) (Fig 2).

A univariate analysis to identify the factors affecting PFS found significant association of PFS with the escalated dose (HR = 0.42, 95%CI: 0.20–1.00) and age (> 70 years: HR = 1.94, 95%CI: 1.07–3.52). In the multivariate analysis, only the escalated dose was identified as a significant predictor for prognosis (HR = 0.48, 95%CI: 0.23–0.98). Using LASSO approach, we found age > 70 (HR = 1.55) and diagnosis year after 2010 (HR = 1.42) were unfavorable predictors of PFS, whereas a larger radiation volume (≥ 250 ml; HR = 0.81) was a favorable predictor (Table 3).

Univariate analyses showed only two factors were significant predictors of OS. The escalated group had a better OS (HR = 0.34, 95%CI: 0.18–0.64), and those who were diagnosed after 2010 had a worse OS (HR = 2.01, 95%CI: 1.12–3.32). The other variables did not have

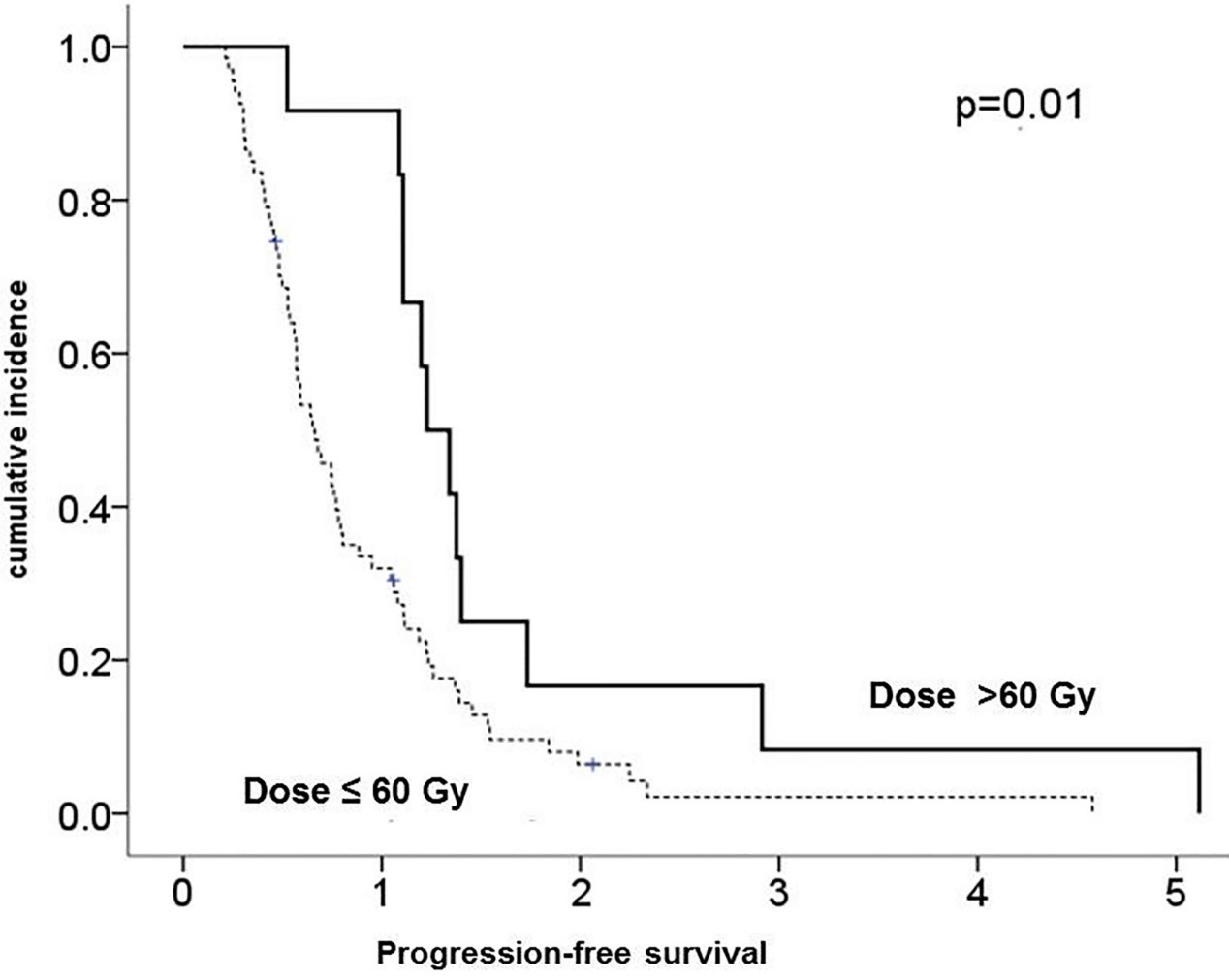

**Fig 1. Kaplan-Meier estimates of progression-free survival comparing glioblastoma patients receiving the standard dose (≤60 Gy) and those receiving dose escalation (> 60 Gy).**

HRs with a p value < 0.10, including age stratification, sex, surgery, chemotherapy, and radiation volume. Further, when the two significant factors (year of diagnosis and radiation dose) were put into the multivariate Cox proportional hazard model, only the escalated dose was identified as an independent predictor for prognosis (HR = 0.40, 95%CI: 0.21–0.78). In addition, the LASSO method selected diagnosis year, radiation volume, and radiation dose into the final model. We also found the escalated dose (HR = 0.47) and a larger radiation volume (HR = 0.76) were predictors for better OS (Table 4). The final model of LASSO based on the criteria of AICC was listed in S1 Table. Following detailed statistical analysis, a moderate radiation dose escalation (> 60 Gy) was found as an independent factor affecting OS in GBM patients.

## Discussion

GBM is the deadliest primary malignant brain tumor, typically presenting as a single brain tumor and tending to recur locally just adjacent to the primary tumor site. Despite

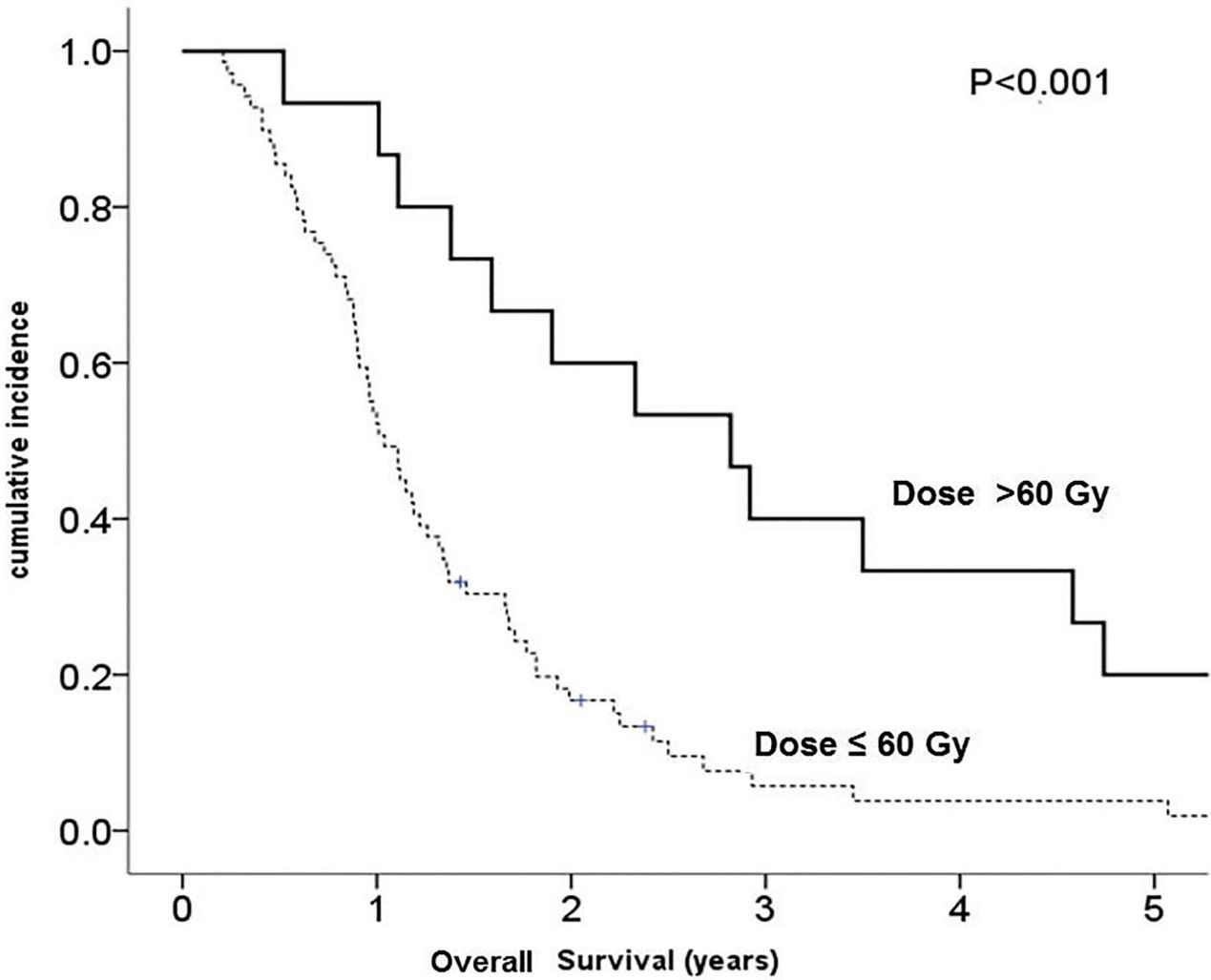

**Fig 2. Kaplan-Meier estimates of overall survival comparing glioblastoma patients receiving a standard dose (≤60 Gy) and those receiving dose escalation (> 60 Gy).**

improvements in imaging, surgery, radiotherapy, and chemotherapy, the majority of GBM patients continue to recur at the site of the original tumor site. According to previous studies on the pattern of failure for GBM patients, more than 80% of tumors recurred within 2 cm of the pre-surgical tumor margin. Local tumor progression or relapse within or adjacent to the original brain lesion occurs in approximately 90% of GBM patients [3–5]. Therefore, local control is most paramount to improving the GBM outcomes. There are many clinical studies on the extent of surgery and escalated radiation dose to prevent local recurrence. However, clinical circumstances are often associated with difficulties in achieving a GTR, such as old age, comorbidities, performance status, large tumor size, and multiple lesions [4, 6, 15–17].

A pooled analysis of six randomized trials of radiotherapy versus no radiotherapy following surgery reported significant survival benefits in the cohort of radiotherapy [18, 19]. However, the survival following standard 60-Gy radiation was poor, and OS remained dismal without long-term survivors. Therefore, prescribing a higher radiation dose for GBM has been attempted in many clinical studies, including altered fractionation, stereotactic radiosurgery,

**Table 3. Univariate and multivariate Cox regression analyses of progression-free survival for glioblastomas.**

| | Univariate | | Multivariate[a] | | Multivariate[b] | |
|---|---|---|---|---|---|---|
| Variable | HR (95% CI) | p-value | HR (95% CI) | p-value | HR(95% CI) | p-value |
| Age group | | | | | | |
| ≤ 70 | (reference) | | | | | |
| > 70 | 1.94 (1.07–3.52) | 0.03 | 1.64 (0.90–3.00) | 0.11 | 1.55(0.78–3.09) | 0.21 |
| Sex | | | | | | |
| female | (reference) | | | | | |
| Male | 1.17 (0.73–1.89) | 0.63 | NA | | | |
| Radiation dose | | | | | | |
| ≤ 60 Gy | (reference) | | | | | |
| > 60 Gy | 0.42 (0.22–0.81) | 0.01 | 0.48 (0.23–0.98) | 0.04 | | |
| Radiation volume[b] | | | | | | |
| < 250ml | (reference) | | | | | |
| ≥ 250ml | 0.78 (0.46–1.34) | 0.38 | NA | | 0.81(0.45–1.39) | 0.44 |
| Surgery[c] | | | | | | |
| No | (reference) | | | | | |
| Biopsy | 0.91 (0.54–1.54) | 0.74 | NA | | | |
| Chemotherapy | | | | | | |
| No | (reference) | | | | | |
| Yes | 0.86 (0.43–1.73) | 0.67 | NA | | | |
| Year of diagnosis | | | | | | |
| 2005–2010 | (reference) | | | | | |
| 2010–2016 | 1.51 (0.93–2.47) | 0.098 | 1.09 (0.64–1.86) | 0.75 | 1.42(0.73–2.76) | 0.30 |

HR, hazard ratio; CI, confidence interval, NA: not available (not included in the model).

[a]p value <0.10 were included in multivariable analysis.

[b]Variables were selected using LASSO method in multivariate analysis.

[c]27 patients did not have detailed radiation volume.

[d]Subtotal or gross-total resection, other than biopsy only.

and brachytherapy to deliver a higher dose to the PTV region with the hope of improving local tumor control [5, 8–10]. A clear dose-response relationship has not been identified for GBM. However, our study supports the survival benefit of GBM using the IMRT technique to deliver a moderately escalated dose (> 60-Gy) with conventional 1.8–2.0 Gy daily fractionation in contrast to standard 60-Gy regimen only. Our finding is well in line with a dose escalation study of Zschaeck et al [11] with 133 patients receiving standard 60-Gy radiotherapy and 23 receiving moderate dose escalation (66-Gy), which found the higher dose lead to decreased intracranial recurrence, i.e. better OS. Zhong et al. also reported improved PFS and OS following a moderately integrated radiation boost scheme (64 Gy over 27 fractions to GTV) in GBM patients [12]. In a study by Graf et al., the median survival time was 3.0 months for a radiation dose of 55 Gy or less, 8.6 months for doses between 56 and 65 Gy, and 9.6 months for a dose range between 66 and 75 Gy ($p < 0.01$) [20]. Two other studies using historical data also showed a survival benefit in patients who received a higher radiation dose [21, 22].

On the contrary, the RTOG 98–03 study recruited GBM patients to receive four escalated radiation dosages (66, 72, 78, and 84 Gy) and found no survival benefits among groups with dose escalation. However, the results concluded the feasibility of delivering dosages higher than the standard 60 Gy with an acceptable toxicity of late brain necrosis [9]. Nakagawa et al. reported a trial of dose escalation up to 90 Gy and found those receiving such a high dose had

**Table 4. Univariate and multivariate Cox regression analyses of overall survival for glioblastomas.**

| Variables | Univariate | | Multivariate[a] | | Multivariate[b] | |
|---|---|---|---|---|---|---|
| | HR (95%CI) | p-value | HR (95%CI) | p-value | HR(95%CI) | p-value |
| Age group | | | | | | |
| ≤ 70 | (reference) | | | | | |
| >70 | 1.29 (0.76–2.20) | 0.35 | NA | | | |
| Sex | | | | | | |
| female | (reference) | | | | | |
| Male | 1.18 (0.74–1.86) | 0.49 | NA | | | |
| Radiation dose | | | | | | |
| ≤ 60 Gy | (reference) | | | | | |
| > 60 Gy | 0.34 (0.18–0.64) | < 0.001 | 0.40 (0.21–0.78) | 0.007 | 0.47(0.17–1.35) | 0.16 |
| Radiation volume[b] | | | | | | |
| < 250 mL | (reference) | | | | | |
| ≥ 250 mL | 0.75 (0.42–1.31) | 0.30 | NA | | 0.76(0.43–1.34) | 0.34 |
| Surgery | | | | | | |
| No | (reference) | | | | | |
| Yes | 0.93 (0.55–1.58) | 0.78 | NA | | | |
| Chemotherapy | | | | | | |
| No | (reference) | | | | | |
| Yes | 0.77 (0.43–1.38) | 0.37 | NA | | | |
| Chemotherapy regimen[e] | | | | | | |
| No chemotherapy | (reference) | | | | | |
| TMZ-based | 0.56 (0.30–1.08) | | | | | |
| Non-TMZ | 0.54 (0.18–1.60) | 0.21 | NA | | | |
| Year of diagnosis | | | | | | |
| 2005–2010 | (reference) | | | | | |
| 2010–2016 | 2.01 (1.12–3.32) | 0.006 | 1.52 (0.9–2.57) | 0.12 | 1.98(0.98–4.02) | 0.06 |

HR, hazard ratio; CI, confidence interval; NA: not available (not included in the model); TMZ, temozolomide.

[a]p value < 0.10 were included in multivariable analysis.

[b]Variables were selected using LASSO method in multivariate analysis.

[c]27 patients did not have detailed radiation volume.

[d]Subtotal or gross-total resection, other than biopsy only.

[e]6 patients did not have detailed records of chemotherapy regimen.

a lower rate of local recurrence, whereas the OS was similar to the conventional dose group, though this study concluded that the benefit of a higher dose was not clear [10]. Additional reports have indicated that an escalated dose beyond 60 Gy for GBM did not improve OS [5, 9, 10, 23]. To conclude, the National Comprehensive Cancer Network still considered 60-Gy radiotherapy as the standard of care. However, the debate of the pros and cons of an escalated radiation dose in GBM have continued. While the optimal method for delivering post-operative dose or methods of radiation remains unclear, NGR BN-001 conducted a phase II trial study to compare hypofractionated dose-escalated photon or proton beam radiation therapy with standard-dose radiation therapy to evaluate dose escalation for GBM patients. As a result of effort to summarize the findings in previous studies, we conducted a literature review to identify the relationship between escalation and outcome. Databases of Medline and PubMed were searched. Clinical cases series and Radiation Therapy and Oncology Group reports are listed in Table 5.

**Table 5. Literature review on the escalated radiation dose trial of glioblastoma.**

| Study [reference] | Escalated radiation dose trial | N | Median survival (month) | Concurrent chemotherapy | Survival benefit |
|---|---|---|---|---|---|
| Nakagawa et al. (1998)[10] | Low dose (60–80Gy) | 21 | 17 | Nimustine, vincristine | No |
| | High dose (90 Gy) | 17 | | | |
| RTOG 98-03(2002) [9] | PTV< 75 cm3 46 Gy in 23 fractions + 20–38 Gy (3DCRT) | 94 | 11.6 (66 Gy) | BCNU | No |
| | | | 11.8 (72 Gy) | | |
| | | | 11.8 (78 Gy) | | |
| | | | 19.3 (84 Gy) | | |
| | PTV≥ 75 cm3 46 Gy in 23 fractions + 20–38 Gy (3DCRT) | 109 | 8.2 (66 Gy) | | |
| | | | 6.5 (72 Gy) | | |
| | | | 6.9 (78 Gy) | | |
| | | | 6.0 (84 Gy | | |
| Chan et al. (2002) [5] | 60 Gy + 10 Gy (3DCRT) | 20 | 13.9 | No | No |
| | 60 Gy + 20 Gy | 55 | 12.9 | | |
| | 60 Gy + 30 Gy | 34 | 11.7 | | |
| RTOG 93-05(2004)[23] | 60 Gy in 30 fractions | 97 | 13.6 | BCNU | No |
| | 60 Gy + 15–24 Gy (stereostatic radiosurgery) | 89 | 13.5 | | |
| Graf et al. (2005) [20] | 56-65Gy | 83 | 8.6 | No | Yes |
| | 66-75Gy | 52 | 9.6 | | |
| Tsien et al. (2012)[22] | 66–81Gy | 38 | 20.1 | TMZ | Yes |
| Zschaeck et al. (2018)[11] | 60Gy | 133 | 15.3 | TMZ | Yes |
| | 66Gy | 23 | 18.8 | | |
| Kim et al. (2019) [21] | 66-81Gy | 82 | 18.7 | TMZ | Yes |
| Zhong et al. (2019) [12] | 64 Gy in 27 fractions (SIB) | 80 | 21 | TMZ | Yes |
| Current study | 60 Gy in 30–33 fractions | 69 | 12.5 | TMZ | Yes |
| | >60 Gy (conventional boost) | 15 | 33.8 | | |

RTOG, radiation therapy and oncology group; SIB, simultaneous integrated boost; 3D CRT, 3D conformal radiotherapy; TMZ: temozolomi

It is well-known that achieving GTR is crucial in the management in GBM. However, GBM is a diffusely infiltrative brain tumor, where tumor cells often extend beyond the maximum resection. Therefore, GTR only should be regarded as a debulking operation, and, in the most clinical scenarios, extensive radical surgery is not feasible. Kreth et al. reported only about 50% of patients could receive GTR and those that did had better survival [16]. Awad et al. found OS significantly improved with the extent of resection in a univariate analysis, averaging 22, 19, and 13 months for > 90%, 80–90%, and 70–80% tumor resection, respectively. However, after multivariate analysis, the extent of resection no longer correlated with OS [24]. Our study only differentiated between tumor resection and biopsy-only from hospital database analysis and found no significant association between resection and survival. Nonetheless, radical surgery should be weighted between the clinical recovery and the risk of neurologic deficit. Although we failed to demonstrate the benefits of the extent of resection in our cohort, we believe the extent of resection is imperative in GBM. Studies supported the resection of more contrast-enhancing portion of a GBM leads to increased survival. Additional resection of the T2-weighted MRI region appears to confer an added survival advantage [24–26]. Radiation volume should be highly correlated with the extent of tumor resection. In our study, the median radiation treatment volume was similar between the two groups (249 ml in dose escalation *vs*. 247 ml in standard dose). A larger radiation volume (≥ 250 ml) did not yield better OS and PFS in the univariate analysis. However, the LASSO selector identified radiation volume ≥ 250ml as a favorable predictor for PFS (HR = 0.81) and OS (HR = 0.76) as well.

We analyzed several prognostic variables including age, sex, and chemotherapy. The date of diagnosis before 2010 was a significant factor for worse OS in the univariate analysis. However, after adjusting for radiation dose, it was no longer significant in the OS multivariate analysis. Whereas diagnosis year before 2010 was not associated with OS or PFS in proportional hazards model, when we conducted Cox models with a LASSO selector, it was included in the models for both OS and PFS, signifying diagnosis year might be a confounding factor. The number of GBM patients assigned to accept dose escalation with conventional boost had decreased after 2010 at our hospitals. The rationale for opting for standard 60-Gy, derived from the concern that either dose intensification may come with a cost, or dose intensification has involved utilizing various techniques, such as hypofractionated or simultaneous integrated boost dose, and/or stereotactic radiosurgery boost [25].

The age of the cohort in the escalated dose group was younger than the cohort in the standard group in our study. Younger age is significantly associated with better survival in GBM patients in the literature [26, 27]. Again, Cox models with a LASSO selector identified age as a potential confounding factor on survival in our study which could impact the results. Evidence-based clinical practice guidelines recommend patients older than 70 years old consider hypofractionated radiotherapy [28]. So, we used 70 years as a cutoff and found the age > 70 group had a worse PFS in the univariate analysis. However, after adjusting for the radiation dose in multivariate analysis, it was no longer a significant predictor for PFS. Roa et al. randomly assigned GBM patients aged 60 years or older to receive either a standard course (60 Gy in 30 fractions over 6 weeks) or a shorter course (40 Gy in 15 fractions over 3 weeks) of radiation after surgery and found no differences in survival between the two groups [28], which is compatible to our finding.

Currently, GBM patients receive adjuvant TMZ as the standard chemotherapy regimen. Nonetheless, only 46% of our patients received TMZ regimen, and chemotherapy dose and regimen did not conform to the standards in our study. The reason was that TMZ was not fully reimbursed by the National Health Insurance in Taiwan in the early 2000s. Therefore, patients were treated with various chemotherapy regimens, like BCNU and cisplatin. Non-uniform chemotherapy might mask the significance of chemotherapy in our study.

The 2016 WHO classification of central nervous tumors introduced molecular parameters in addition to histology to define GBM tumor entities [29]. However, we enrolled GBM patients between 2005 and 2016, and IDH and MGMT markers were not mandatory for routine pathologic reports. Because of the lack of data, we were unable to evaluate the prognostic significance of these molecular markers in our study.

## Conclusions

Although the current study was limited by its retrospective design, the escalated dose (> 60-Gy) with similar treatment volumes resulted in survival benefit after adjusting for other prognostic variables. Dose escalation was still identified as an independent factor for OS by using Cox models with a LASSO selector. The limitations of this study include non-uniform chemotherapeutic treatment, longer accrual of patients, younger age in the high dose group, and those along with the general restrictions of a retrospective analysis. In addition, the small cohort and relatively large number of factors of interest limited the potential for multiple comparisons and the evaluation of certain factors, such as completeness of resection. Moreover, the molecular characteristics, such as MGMT or IDH mutation status, were not routinely assessed in the cohort.

## Supporting information

**S1 Table. Final models using the least absolute shrinkage and selection operator method.**
(DOCX)

**S1 Data.**
(XLSX)

## Acknowledgments

The authors thank the Department of Cancer Registry of Chi Mei Medical Center, Tainan, Taiwan and Chi-Mei Hospital, Liouying, Tainan, Taiwan for their assistance to this study.

## Author Contributions

**Conceptualization:** Li-Tsun Shieh, Sheng-Yow Ho.

**Data curation:** Li-Tsun Shieh.

**Funding acquisition:** Sheng-Yow Ho.

**Methodology:** How-Ran Guo, Chung-Han Ho, Sheng-Yow Ho.

**Project administration:** Sheng-Yow Ho.

**Resources:** Li-Ching Lin, Chin-Hong Chang.

**Writing – original draft:** Li-Tsun Shieh.

**Writing – review & editing:** How-Ran Guo, Sheng-Yow Ho.

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
