## [Decision Letter · Decision Letter 0]

3 Mar 2020

PONE-D-20-02836

Survival of glioblastoma treated with a moderately escalated radiation dose – Results of a retrospective analysis

PLOS ONE

Dear Dr. Ho,

Thank you for submitting your manuscript to PLOS ONE. After careful consideration, we feel that it has merit but does not fully meet PLOS ONE’s publication criteria as it currently stands. Therefore, we invite you to submit a revised version of the manuscript that addresses the points raised during the review process.

We would appreciate receiving your revised manuscript by Apr 17 2020 11:59PM. To enhance the reproducibility of your results, we recommend that if applicable you deposit your laboratory protocols in protocols.io, where a protocol can be assigned its own identifier (DOI) such that it can be cited independently in the future. For instructions see: http://journals.plos.org/plosone/s/submission-guidelines#loc-laboratory-protocols

We look forward to receiving your revised manuscript.

Kind regards,

Stephen Chun

Academic Editor

PLOS ONE

Journal Requirements:

2. - In the ethics statement in the manuscript and in the online submission form, please provide additional information about the patient records used in your retrospective study. Specifically, please ensure that you have discussed whether all data were fully anonymized before you accessed them and/or whether the IRB or ethics committee waived the requirement for informed consent. If patients provided informed written consent to have data from their medical records used in research, please include this information.

- Thank you for providing the following information about study approval in the Methods: "This study was approved by our institutional review board and ethics committee." Please revise this statement to include the full name of the Institutional Review Board and Ethics Committee that approved the study, and provide the approval/permit number that was issued on approval.

- In the Methods, please provide additional information about the literature review, including 1) the keywords that were used to retrieve the search results, 2) the date(s) on which the search was performed, and 3) details of any analysis of study quality, heterogeneity or bias that was performed.

Reviewers' comments:

Reviewer's Responses to Questions

**Comments to the Author**

1. Is the manuscript technically sound, and do the data support the conclusions?

Reviewer #1: Partly

Reviewer #2: Partly

Reviewer #3: Yes

2. Has the statistical analysis been performed appropriately and rigorously? 

Reviewer #1: Yes

Reviewer #2: Yes

Reviewer #3: Yes

3. Have the authors made all data underlying the findings in their manuscript fully available?

Reviewer #1: Yes

Reviewer #2: Yes

Reviewer #3: Yes

4. Is the manuscript presented in an intelligible fashion and written in standard English?

Reviewer #1: Yes

Reviewer #2: Yes

Reviewer #3: No

5. Review Comments to the Author

Reviewer #1: I commend the authors for their endeavors in investigating a question of clinical significance/interest for a malignancy that despite recent improvements (TTF, etc) remains devastating.

There are a few areas that I believe could be modified which would substantially strengthen this manuscript

1) While the authors acknowledge the lack of molecular characterization, the absence of capturing/adjusting for IDH/MGMT severely limits interpretation of clinical outcomes in a small cohort. If at all possible I would encourage the authors to adjust for either of the above even if many values are missing.

2) I would encourage the authors to consider using statistical modeling such as penalized Cox models with a LASSO selector given the small cohort and relatively large number of factors of interest. More robust methodology would further limit the potential for multiple comparison error.

3) Rather than adjusting for chemotherapy/non-chemotherapy, I would encourage the authors to adjust for temozolomide/non-tmz chemo/none given the substantial improvement with TMZ in MGMT methylated patients (perhaps to some extent in unmethylated patients as well given the potential for protein expression despite methylation status).

4) Would recommend the authors mention NRG-BN-001 which is investigating dose-escalated RT for patients with GBM in the era of TMZ.

5) It is a little concerning to me outcomes were worse for patients diagnosed after 2010. Given multiple large database analyses have associated improved OS in more modern cohorts (PMID 27214765, 28452053) this finding to me highlights the potential presence of an unadjusted maldistribution of molecular mutations (IDH/MGMT) which may be modulating PFS/OS.

Reviewer #2: This retrospective study evaluated the effect of radiation dose on survival outcomes for patients with glioblastoma and identified dose escalation as being associated with improved outcomes. The authors' study adds to the literature regarding dose escalation. Since GBM frequently recurs locally, it is important to identify factors that correlate with local recurrence. Although the dose escalation group only included 15 patients, it is interesting that these patients did experience improved OS. Only one patient in this group was over the age of 70, while there were 18 patients over age 70 in the standard dose arm. Although age was found to not be associated with outcomes on multivariate analysis, it is probable that this would be significant with a larger sample size since age is so critical to GBM outcomes. One recommendation for the authors is to more clearly separate which patients underwent GTR or STR, since completeness of resection is also associated with improved outcomes in this patient population. Additionally, I recommend more clearly highlighting that there was no difference in treatment volume between the two arms. Overall, this is an interesting study, and although retrospective in nature with all the requisite caveats and without molecular data, it adds an important data point to GBM literature with respect to radiation dose.

Reviewer #3: Overall the manuscript is logical and states its purpose. The subject is not original, but the data and the patients cohorts reported in the study are unique, thereby meeting one of the criteria for publishing in PLOS One. It would be good if there were more patients, and I would encourage the authors to see if they can combine patient cohorts with other centers, recognizing that this may not ultimately be possible. There are, however, many grammatical and work usage errors in the manuscript, and at a minimum these need to be corrected. I am scanning the manuscript with my copyedits in blue.

6. PLOS authors have the option to publish the peer review history of their article (what does this mean?). If published, this will include your full peer review and any attached files.

Reviewer #1: No

Reviewer #2: No

Reviewer #3: No

---

## [Author Response · Author response to Decision Letter 0]

15 Apr 2020

Response to Reviewers’ Comments

First, we feel very happy to receive substantive and excellent comments from reviewers. We respond the comments line by line accordingly. The revised manuscript was marked in red in corrected words, paragraphs, and sections. 

Comments from Reviewer 1：

1. While the authors acknowledge the lack of molecular characterization, the absence of capturing/adjusting for IDH/MGMT severely limits interpretation of clinical outcomes in a small cohort. If at all possible I would encourage the authors to adjust for either of the above even if many values are missing. 

Response: 

The 2016 WHO classification of central nervous tumors has introduced molecular parameters in addition to histology to define GBM tumor entities. However, we enrolled GBM patients between 2005 and 2016, and so the markers of IDH mutation and MGMT were not mandatory in routine pathologic reports. Actually, in our cohort, we could not find any one who had been checked for the status of MGMT. In response to the Reviewer’s comment, we changed the last sentence of Discussion to “Because of the lack of data, we were unable to evaluate the prognostic significance of these molecular markers in our study.” in the revised manuscript. We have also added this as a limitation to Discussion and Conclusion in the revised manuscript. 

 2. I would encourage the authors to consider using statistical modeling such as penalized Cox models with a LASSO selector given the small cohort and relatively large number of factors of interest. More robust methodology would further limit the potential for multiple comparison error. 

Response: 

According to the Reviewer’s suggestion, we performed the analysis and added the results of LASSO approach in Table 3, Table 4, and Supplement Table 1. Following detailed statistical analysis, a moderate radiation dose escalation (> 60 Gy) was still identified as an independent factor affecting OS in GBM patients. In addition, we have added related description to the Abstract and main text in the revised manuscript. 

3. Rather than adjusting for chemotherapy/non-chemotherapy, I would encourage the authors to adjust for temozolomide/non-tmz chemo/none given the substantial improvement with TMZ in MGMT methylated patients (perhaps to some extent in unmethylated patients as well given the potential for protein expression despite methylation status). 

Response: 

Among the 69 (82%) patients who received chemotherapy, only 46 (67% of the total) were prescribed with a TMZ-based regimen. In response to the Reviewer’s comments, we have tried to make comparison among three groups, namely temozolomide chemotherapy, non-TMZ chemotherapy, and no chemotherapy, but found no significant differences regarding overall survival (OS). We have added the result in Table 4.The results are shown in following table:

 Univariable (OS) 

Variable HR (95% CI) p-value

No chemotherapy (reference) 0.211

TMZ-based regimen 0.56(0.30-1.08) 

Non-TMZ Chemotherapy 0.54(0.18-1.60) 

4. Would recommend the authors mention NRG-BN-001 which is investigating dose-escalated RT for patients with GBM in the era of TMZ. 

Response: 

According to the Reviewer’s suggestion, we have added related information to Discussion in the revised manuscript. 

5. It is a little concerning to me outcomes were worse for patients diagnosed after 2010. Given multiple large database analyses have associated improved OS in more modern cohorts (PMID 27214765, 28452053) this finding to me highlights the potential presence of an unadjusted maldistribution of molecular mutations (IDH/MGMT) which may be modulating PFS/OS. 

Response: 

We can understand the Reviewer’s concern that the survival became worse in patients who received treatment in recent years. Although we have no data to address the possible effects of IDH/MGMT, we believe it was quite likely attributable to dose escalation. Most of the patients who received the escalated dose were diagnosed before 2010, while most of those who received the standard dose were diagnosis after 2010. We have added related description to Discussion in the revised manuscript. 

Comment from Reviewer 2：

This retrospective study evaluated the effect of radiation dose on survival outcomes for patients with glioblastoma and identified dose escalation as being associated with improved outcomes. The authors' study adds to the literature regarding dose escalation. Since GBM frequently recurs locally, it is important to identify factors that correlate with local recurrence. Although the dose escalation group only included 15 patients, it is interesting that these patients did experience improved OS. Only one patient in this group was over the age of 70, while there were 18 patients over age 70 in the standard dose arm. Although age was found to not be associated with outcomes on multivariate analysis, it is probable that this would be significant with a larger sample size since age is so critical to GBM outcomes. One recommendation for the authors is to more clearly separate which patients underwent GTR or STR, since completeness of resection is also associated with improved outcomes in this patient population. Additionally, I recommend more clearly highlighting that there was no difference in treatment volume between the two arms. Overall, this is an interesting study, and although retrospective in nature with all the requisite caveats and without molecular data, it adds an important data point to GBM literature with respect to radiation dose.

Response: 

We thank the Reviewer for the encouraging comment. While we agree that the completeness of resection is also associated with improved outcomes, because of the small case number, we would not have the statistical power to adjust for more variables. We added “the completeness of resection is also associated with improved outcomes, because of the small case number, we could not evaluate or adjust its effects in our study” as a limitation to both Discussion. As suggested by the Reviewer, we also added “but their treatment volumes were similar to the others” to the Abstract and “with similar treatment volumes” to the Discussion to highlight that there was no difference in treatment volume between the two arms.

Comment from Reviewer 3：

Overall the manuscript is logical and states its purpose. The subject is not original, but the data and the patients cohorts reported in the study are unique, thereby meeting one of the criteria for publishing in PLOS One. It would be good if there were more patients, and I would encourage the authors to see if they can combine patient cohorts with other centers, recognizing that this may not ultimately be possible. There are, however, many grammatical and work usage errors in the manuscript, and at a minimum these need to be corrected. I am scanning the manuscript with my copyedits in blue. 

Response:

We agree with the Reviewer’s suggestion that a multicenter study is a good idea to expand the database, but it is not achievable in a short period time. We believe the publication of this manuscript will provide a more solid ground to call for such a study. We really appreciate the editing of the manuscript and have corrected those and other errors in the manuscript.

---

## [Decision Letter · Decision Letter 1]

30 Apr 2020

Survival of glioblastoma treated with a moderately escalated radiation dose – Results of a retrospective analysis

PONE-D-20-02836R1

Dear Dr. Ho,

We are pleased to inform you that your manuscript has been judged scientifically suitable for publication and will be formally accepted for publication once it complies with all outstanding technical requirements.

With kind regards,

Stephen Chun

Academic Editor

PLOS ONE

Additional Editor Comments (optional):

Reviewers' comments:

Reviewer's Responses to Questions

**Comments to the Author**

1. If the authors have adequately addressed your comments raised in a previous round of review and you feel that this manuscript is now acceptable for publication, you may indicate that here to bypass the “Comments to the Author” section, enter your conflict of interest statement in the “Confidential to Editor” section, and submit your "Accept" recommendation.

Reviewer #2: All comments have been addressed

Reviewer #3: All comments have been addressed

2. Is the manuscript technically sound, and do the data support the conclusions?

Reviewer #2: Yes

Reviewer #3: (No Response)

3. Has the statistical analysis been performed appropriately and rigorously? 

Reviewer #2: Yes

Reviewer #3: (No Response)

4. Have the authors made all data underlying the findings in their manuscript fully available?

Reviewer #2: Yes

Reviewer #3: (No Response)

5. Is the manuscript presented in an intelligible fashion and written in standard English?

Reviewer #2: Yes

Reviewer #3: (No Response)

6. Review Comments to the Author

Reviewer #2: The authors have appropriately addressed reviewer concerns and I recommend the manuscript is accepted for publication.

Reviewer #3: (No Response)

7. PLOS authors have the option to publish the peer review history of their article (what does this mean?). If published, this will include your full peer review and any attached files.

Reviewer #2: No

Reviewer #3: No

---

## [Editor Report · Acceptance letter]

4 May 2020

PONE-D-20-02836R1 

Survival of glioblastoma treated with a moderately escalated radiation dose – Results of a retrospective analysis 

Dear Dr. Ho:

I am pleased to inform you that your manuscript has been deemed suitable for publication in PLOS ONE. Congratulations! Your manuscript is now with our production department. 

With kind regards,

on behalf of

Dr. Stephen Chun 

Academic Editor

PLOS ONE